# Characterization of *Salmonella* spp. and *E. coli* Strains Isolated from Wild Carnivores in Janos Biosphere Reserve, Mexico

**DOI:** 10.3390/ani12091064

**Published:** 2022-04-20

**Authors:** Jonathan J. López-Islas, Estela T. Méndez-Olvera, Daniel Martínez-Gómez, Andrés M. López-Pérez, Libertad Orozco, Gerardo Suzan, Carlos Eslava

**Affiliations:** 1Doctorado en Ciencias Agropecuarias, Universidad Autónoma Metropolitana, Calz. del Hueso1100, Villa Quietud, Coyoacán, Ciudad de México City 04960, Mexico; jonathan.66mvz@gmail.com; 2Departamento de Producción Agrícola y Animal, Universidad Autónoma Metropolitana, Calz. del Hueso 1100, Villa Quietud, Coyoacán, Ciudad de México City 04960, Mexico; 3Departamento de Etología, Fauna Silvestre y Animales de Laboratorio, Facultad de Medicina Veterinaria y Zootecnia, Universidad Nacional Autónoma de México, Avenida Universidad 3000, Ciudad de México City 04510, Mexico; amlope@ucdavis.edu (A.M.L.-P.); gerardosuz@gmail.com (G.S.); 4Fundación para el Manejo y la Conservación de la Vida Silvestre FMCOVIS A.C., Avenida Universidad 3000, Ciudad Universitaria, Ciudad de México City 04510, Mexico; libertad.orozco@hotmail.com; 5Unidad Periférica Investigación Básica y Clínica de Enfermedades Infecciosas-Hospital Infantil de México Federico Gómez, Facultad de Medicina, Universidad Nacional Autónoma de México, Ciudad de México City 04510, Mexico; carlos_01eslava@yahoo.com.mx

**Keywords:** *Salmonella*, *Escherichia coli*, zoonoses

## Abstract

**Simple Summary:**

Emerging diseases (EIDs) represent a constant challenge in public health. With the recent emergence of new pathogens, some questions about the mechanisms and sites where they are generated have aroused interest. Natural environments could be the sites where pathogenic microorganisms find the conditions to generate new variants. It has been established that approximately 60.3% of EIDs are caused by potentially zoonotic pathogens, of which more than half are thought to have originated from wild individuals. In this aspect, carnivores can play an important role in the dynamics of various diseases, since there are species that are widely distributed, roam large areas, and can be carriers of a wide range of microorganisms, some of which are zoonotic. The results obtained in this work show that different species of wild carnivores can be carriers of atypical strains of pathogenic microorganisms, which shows that natural environments can represent important sites for the study of EIDs.

**Abstract:**

Enterobacteriaceae are considered one the most important zoonotic pathogens. In this study, we analyzed the characteristics of *E. coli* and *Salmonella* spp. strains present in carnivores from Janos Biosphere Reserve, Mexico. These microorganisms had been isolated from a wide range of domestic and free-range animals, including wild carnivores. Fifty-five individuals were sampled, and the presence of *Salmonella* and *E. coli* was determined by bacteriological standard methods. Strains isolated were characterized by molecular methods and in vitro infection assays. Eight different species of carnivores were captured, including coyotes (*Canis latrans*), gray fox (*Urocyon cinereoargenteus*), desert foxes (*Vulpes macrotis*), striped skunks (*Mephitis mephitis*), hooded skunks (*Mephitis macroura*), lynxes (*Lynx rufus*), raccoons (*Procyon lotor*), and badgers (*Taxidea taxus*). *Salmonella* spp. and *E. coli* were isolated from four species of carnivores. Five *Salmonella* spp. strains were isolated, and their molecular characterization revealed in three of them the presence of fimbrial and virulence genes associated with cell invasion. In vitro evaluation of these strains showed their capability to invade human Hep2 cells. Sixty-one *E. coli* strains were isolated; different serotypes and phylogroups were observed from these strains. Additionally, the presence of virulence genes showed differently.

## 1. Introduction

Emerging infectious diseases (EIDs) are defined as diseases that have recently increased in incidence or geographic range, recently moved into new host populations, recently been discovered, or are caused by newly evolved pathogens [1]. EIDs are a significant problem for public health and global economies. Their emergence is thought to be driven largely by environmental, ecological, and anthropogenic factors s [2]. General strategies to reduce the negative effects of emerging diseases are focused on post-emergence outbreak control, through quarantine procedures, antimicrobial or antiviral drug development, and vaccination. However, delays in recognition of new pathogens in natural areas, lack of scientific research about the genetic characteristics, combined with increased urbanization and global connectivity, have resulted in recent EIDs causing extensive worldwide mortality and important economic damage (e.g., SARS, EHEC O104:H4) [3].

Wildlife populations are considered play an important role in pathogen emergence, by forming the reservoirs from which zoonotic pathogens (pathogens transmissible between animals and humans) may emerge [1]. Enteric infections are one of the most important causes of morbidity and mortality throughout the world, particularly in children. In recent years, scientific research, focused on pathogenesis, treatment development, and prevention strategies, has made important progress. Unfortunately, enteric infections remain an important public health worldwide problem [4]. New outbreaks of enteric infections produced by *Salmonella* spp., *Campylobacter* spp., and Shiga toxin-producing *Escherichia coli* (STEC) are reported each year, with new virulence properties and epidemiological characteristics. This phenomenon showed the capability of pathogen agents to evolve and develop mechanisms for spreading to and infecting new hosts, and therefore represent a public health challenge [5].

*Escherichia coli* is the most abundant bacteria in the intestines of mammals including humans, and diverse microbiome studies showed their abundance in many species of animals. New pathogenic *E. coli* strains have been isolated and described in recent years; understanding the origin of these pathogenic strains is important for developing control strategies [6,7]. A significant number of acute *E. coli* infections are known to have zoonotic origins. Diverse studies have shown that diverse pathogenic *E. coli* strains are present in natural areas and nonhuman hosts, representing reservoirs for pathogenic strains [8]. *Salmonella* spp. represents an important zoonotic enteropathogen that affects humans, livestock, wildlife, companion, and zoo animals [9]. Although the intestinal tract of production animals is the primary reservoir for non-typhoidal *Salmonella*, it is also found in many wildlife species [10]. Strains reported in wildlife have been isolated from humans, suggesting that wildlife species can be reservoirs for *Salmonella* spp. [11]. In this study, different *E. coli* and *Salmonella* spp. strains were isolated from a wild population of carnivores from the Janos Biosphere Reserve Chihuahua, Mexico, and the virulence characteristics of strains were described.

## 2. Materials and Methods

### 2.1. Study Area

This study was performed in the Janos Biosphere Reserve (RBJ) in Chihuahua Mexico, at the northwest of Chihuahua, located between meridians 108°56′49″ and 108°56′22″ W and parallels 31°11′7″ and 30°11′27″ N. The reserve has an area of 5264 km^2^ with an altitude range of 1200–2700 m above sea level. Five sampling locations were selected: “El Cuervo” (30°41′19″ N, 108°15′05″ W), “Ejido San Pedro” (30°53′08″ N, 108°25′57″ W), “Monte Verde” (30°58′28″ N, 108°42′40″ W), “Rancho ojitos” (30°46′52″ N, 108°32′14″ W), and “Rancho San Pedro” (30°41′12″ N, 108°32′46″ W) (Appendix A).

### 2.2. Experimental Design and Animal Management

Wild carnivores were captured at each of the five locations over nine consecutive days per season (autumn and spring), with 16 trapping sets placed at intervals of 500–800 m along a 10 km transect. Each set contained one box trap (Tomahawk Live Trap Inc., Hazelhurst, WI, USA, 76 × 76 × 178 cm or 152 × 51 × 71 cm) and one leg-hold trap (#1.75 or #3 Victor Coil Soft Catch) with at least 30 m distance between them. Each wild carnivore was chemically immobilized with a mixture of ketamine hydrochloride and xylazine hydrochloride (Pisa, México) according to standard doses [12]. Samples of feces were taken through rectal swabs; each swab was placed in a transport medium and kept at 4 °C until processing in the laboratory.

### 2.3. Isolation and Identification of Salmonella spp. from feces

Feces samples collected from carnivores were inoculated on 10 mL of peptone broth (Oxoid, Monterrey, Mexico) and incubated at 37 °C for 18 h. After, 1 mL was transferred to 10 mL of selenite broth (Merck, Branchburg, NJ, USA) and incubated at 37 °C for 20 h. Each selenite broth was seeded on Salmonella-Shigella agar (Merck, Branchburg, NJ, USA) and MacConkey agar (Merck, Branchburg, NJ, USA) and incubated for 24 h at 37 °C. All colonies with characteristic morphology of *Salmonella* spp. were transferred to Salmonella-Shigella agar and incubated at 37 °C for 24 h. Finally, microorganisms were identified by standard microbiology methods. *Salmonella enterica* Typhimurium ATCC 14028 was included in all bacteriological procedures, PCR, and invasion assays, as a reference strain.

### 2.4. Isolation and Identification of E. coli from Feces

Fecal samples were inoculated into 9 mL buffered peptone water and incubated for 24 h at 37 °C; then 0.1 mL was picked and inoculated onto EMB agar (Oxoid, Monterrey, Mexico) and incubated for 24 h at 37 °C. Thirty-six Green metallic sheen-like colonies from each sample were picked and tested in MRVP broth (Oxoid, Monterrey, Mexico), Simmons citrate (Oxoid, Monterrey, Mexico), SIM agar (Oxoid, Monterrey, Mexico), TSI agar (Oxoid, Monterrey, Mexico), and urea agar (Oxoid, Monterrey, Mexico). Isolates that were positive to the methyl red test (Sigma-Aldrich, St. Louis, MO, USA), negative to the Voges–Prokeur test (Merck, Branchburg, NJ, USA), negative to the Simmons citrate test, negative to the H_2_S test, positive to the indole test (Merck, Branchburg, NJ, USA), positive to the motility test, negative to the urease production test, and produced acid over acid reaction in the TSI test, were subcultured on EMB agar (Oxoid, Monterrey, Mexico) at 37 °C for 24 h. One pure culture colony was picked using a sterile loop and suspended in 0.1 mL of LB broth (Merck, Branchburg, NJ, USA) for PCR identification. Six *E. coli* strains, EPEC O127:H6 (E2348/69), EHEC O157:H7 (EOL933), ETEC O78:H11 (H10407), EAEC O42: NM, EIEC O143: NM and UPEC O73: NM., were used as controls in this study. The EPEC O127:H6, EHEC O157:H7, and ETEC O78:H11 are ATCC strains. The EAEC, EIEC, and UPEC (CFT) strains came from human patients with enteric processes, provided by Dr. Carlos A. Eslava Campos, from the Federico Gómez Children’s Hospital.

### 2.5. E. coli and Salmonella spp. Characterization by PCR

All isolates and control strains of *Salmonella* spp. and *E. coli* were seeded in 10 mL of Brain-Heart Infusion broth (Merck, Branchburg, NJ, USA) and incubated for 24 h at 37 °C. The biomass was recuperated by centrifugation at 1200× *g* for 15 min at 4 °C. For DNA extraction, the CTAB-NaCl method was used [13]. The DNA was used for amplification of virulence and genotyping genes by PCR. A commercial kit (PCR Master Mix 2X, Thermo Scientific, Waltham, MA, USA) was used for the PCR assays, following the manufacturers protocol. Primer sequences and PCR conditions are shown in Appendix A. PCR products were analyzed by 1.5% agarose gel electrophoresis.

### 2.6. E. coli Serotyping and Genotyping

Strains identified as *E. coli* were serotyped by agglutination assays using 96-well micro titer plates and rabbit antisera against O1 to O187 somatic (O) antigens and 53 flagellar (H) antigens prepared in rabbits (SERUNAM, registered trademark in Mexico, with number 323158/2015) using a method previously reported [14] with minor modifications. Phylogenetic groups were determined by PCR multiplex using the primers described in Appendix A and under conditions detailed by a method previously reported [15]. The designation of the phylogenetic groups (A, B1, B2, C, D, E, F, and *Escherichia* cryptic clade I) was established by the presence or absence of *chu*A, *yja*A, TspE4.C2, and the *arp*A gene [15].

### 2.7. Salmonella spp. Intracellular Survival Assays

Three independent trials were performed with three replicates at a time, according to the standard protocol for intracellular survival with gentamicin, with some modifications. First, all *Salmonella* strains were grown on Luria Bertani broth supplemented with 0.5% of NaCl for 24 h and harvested in PBS. Bacteria growth was quantified by spectrophotometry and diluted in DMEM (Sigma-Aldrich, St. Louis, MO, USA) at 2 × 10^6^ CFU/mL to prepare inoculum. Hep-2 cells were seeded at 40,000 cells per well in a 48-well dish. After washing three times with PBS, epithelial cells were infected at a multiplicity of infection (MOI) of 100 CFU/cell for 2 h at 37 °C. Following incubation, the monolayers were washed three times with PBS and incubated with DMEM supplemented with 10% fetal bovine serum (Sigma-Aldrich, St. Louis, MO, USA), 50 mM HEPES, and 30 µg/mL gentamicinb (Sigma-Aldrich, St. Louis, MO, USA). At 0, 2, 4, 10, and 24 h post-infection (p.i.), media was removed, and intracellular bacteria were recovered by 1 mL of 1% Triton X-100 (Sigma-Aldrich, St. Louis, MO, USA) addition. The cell lysate was homogenized and transferred to microtubes. In total, 50 µL of each well were diluted and seeded on LB agar for 37 °C for 24 h.

### 2.8. E. coli Adhesion Assays

The adherence assays were carried out using glass coverslips contained in 24-well microplates at 80% cell confluence as previously described [16]. Hep2 cells were grown on DMEM media (Sigma-Aldrich, St. Louis, MO, USA) supplemented with 10% fetal sera (Sigma-Aldrich, St. Louis, MO, USA), and adhesion assays were carried in DMEM media without sera. *E. coli* were grown overnight to an OD600 of 0.8–1.2. Cells were inoculated with an infection dose of 1000:1 bacterium per cell and incubated at 37 °C for five hours. The DMEM medium was removed, and the cells were washed three times with PBS. After, 70% methanol was added for 7 min and cells were stained with Giemsa (Sigma-Aldrich, St. Louis, MO, USA) (10% solution in ddH_2_O) for 40 min at room temperature. Slides were evaluated for adhering bacteria by light microscopy. All tests were repeated at least three times in triplicates. Two distinct patterns of adherence were considered: localized adherence, when the bacteria attached to confined areas of the HEp-2 cells in culture, and diffuse adhesion (DA), when bacteria adhered to the entire surface of the HEp-2.

### 2.9. Statistical Analysis

Descriptive and comparative statistics were performed using R software. *Salmonella* spp. and *E. coli* frequencies by species were established. An analysis of variance (ANOVA) was performed to confirm the significant differences between the means of each infection time in invasion assays.

## 3. Results

### 3.1. Animals Captured

During autumn and spring, a total of 55 carnivores were captured. In autumn, 24 specimens were captured: 7 coyotes (*Canis latrans*), 1 gray fox (*Urocyon cinereoargenteus*), 5 desert foxes (*Vulpes macrotis*), 2 striped skunks (*Mephitis mephitis*), 3 hooded skunks (*Mephitis macroura*), 4 lynxes (*Lynx rufus*), 1 raccoon (*Procyon lotor*), and 1 badger (*Taxidea taxus*). In spring, 31 individuals were captured: 8 coyotes, 9 desert foxes, 5 gray foxes, 1 lynx, 3 raccoons, and 5 badgers. Coyote and desert foxes represented the species with the highest capture frequency, with 27.2% and 22.7%, respectively (Table 1).

### 3.2. Isolation and Identification of E. coli and Salmonella spp. from Feces

To isolate *Salmonella* spp. and *E. coli* in the carnivore population of RBJ, a standard bacteriological method was used. From 55 samples of feces recovered through rectal swabs, five strains of *Salmonella* spp. were isolated (9.09%). The *Salmonella* isolates were retrieved from 1 lynx (*Lynx rufus*), 1 striped skunk (*Mephitis mephitis*), 1 coyote (*Canis latrans*), and 2 gray foxes (*Urocyon cinereoargenteus*) (Table 1).

From the same 55 samples of feces, 1980 strains of *E. coli* were isolated. From these strains, 61 showed positive PCR results for at least one virulence gene evaluated in this study (Appendix A). The *E. coli* isolates were retrieved from 3 coyotes (*Canis latrans*), 2 desert foxes (*Vulpes macrotis*), and 2 striped skunks (*Mephitis mephitis*) (Table 1). It is important to clarify that in this part, only animals with isolations of strains with virulence genes are mentioned. The isolation of commensal *E. coli* was achieved from all individuals.

All *Salmonella* isolates were characterized by PCR to describe the presence of virulence genes (Table 2). In this analysis, a 16s-ITS set of primers was included to confirm bacteriological identification. All *Salmonella* strains showed positive results for 16s-ITS, PCR, but only three strains had positive results for *inv*A, *cdt*B, and *sop*E genes. These strains were isolated from a striped skunk (*Mephitis mephitis*) and 2 gray foxes (*Urocyon cinereoargenteus*). Considering that *inv*A and *sop*E genes are essential virulence factors associated with cell invasion [17,18], strains with negative results for these genes were not included in the analysis of fimbrial operons and invasion assays (Table 2). For PCR analysis of fimbrial operons, the strain 1-B-Uro showed positive results for all evaluated genes. On the other hand, strains 1-BR-Uro and Meph 18 lacked *fim*A and *ste*A genes, and the strain Meph 18 was also deficient in the *stf*A gene (Table 2).

Serotyping of the 61 isolations identified as *E. coli* strains indicated that 46 (75.4%) belonged to different, recognized *E. coli* serogroups and 15 (24.6%) were non-typable (O?). Of the strains with a recognized O group, 17 (36.9%) belonged to O153 pathogenic serogroups, 5 (8.19%) to O139 serogroups, and 4 (6.55%) to O21 serogroup (Table 3 and Appendix A). Most of the *E. coli* strains were included in an unknown group (63.9%), with only 31.14% belonging to virulent groups B2, D, and E (Table 3). The analysis showed that only 3 strains (4.9%) belonged to commensal group A (Table 3).

Appendix A shows the results obtained from genetic evaluation of pathogenic characteristics of the 61 *E. coli* strains isolated from carnivores. The presence of *stx*1 and *stx*2 genes was observed in 25 (40.98%) *E. coli* strains and of these, only 5 (8.19%) showed the presence of both genes, while the remaining 20 strains (32.78%) were positive for only one gene; *stx*1 (3 strains) and *stx*2 (17 strains). The presence of the *esc*V and *eae* genes was observed in 36 strains (59.02%) and of these, 7 strains (11.47) were positive for both genes. These strains were evaluated in cell culture to evaluate their capability to produce localized adhesion (Figure 1). Twenty-three *E. coli* strains were positive for the *esc*V gene, and six for the *eae* gene. The *esc*V-*eae* positive *E. coli* strains belonged to different serotypes, O153H17 and O63H6, and different phylogroups, E, B2, and unknown. For *stx*1-*stx*2 positive strains the serotypes observed were O75H38, O17H18, and O110H28, and all belonged to the unknown phylogroup.

### 3.3. Salmonella spp. Intracellular Survival Assays

*Salmonella* strains showed different capabilities to invade and replicate inside of Hep-2 epithelial cells on intracellular survival assays. In these assays, strain Meph 18 showed the highest ability to replicate inside epithelial cells, even more than the reference strain included in this analysis (Figure 2). At 2 h post-infection, the number of intracellular microorganisms from strain Meph 18 was 5.3 × 10^6^ CFU/mL, in comparison with 3.39 × 10^6^ CFU/mL of *Salmonella* Typhimurium ATCC14028. Meanwhile, for 1-B-Uro and 1-BR-Uro, the number of intracellular microorganisms at this same time were 3.38 and 1.44 × 10^6^ CFU/mL, respectively. The strain 1-BR-Uro showed a reduced ability to replicate inside of epithelial cells (Figure 2). After 4 h post-infection, all strains showed a reduction in the number of CFU/mL, but this could result from loss of cell monolayer. At 24 h post-infection, more than 80% of the cells were lysed by *Salmonella*, compared with 4 h post-infection, where more than 85% of the monolayer was present (data not shown). The results obtained in intracellular survival assays demonstrated that *Salmonella* strains isolated from carnivores were virulent for human cells at different levels.

### 3.4. E. coli Adhesion Assays

Figure 1 shows the occurrence of localized adhesion (LA) in *E. coli* strains isolated from feces. LA occurred with the seven *E. coli* strains with positive PCR results for *eae* and *esc*V evaluated in this study. The results of the adhesion assays showed that at 1:40 h post-infection, all strains showed a LA pattern. Modifications in the cell membrane with a pedestal shape were also observed at 2:30 h postinfection. In the control strain, AL was observed at 1:40 h and the pedestal-like structure at 2:30 h postinfection.

## 4. Discussion

Identification of animal reservoirs for bacterial pathogens are important for understanding and preventing disease emergence. *Salmonella* spp. and *E. coli* are important pathogens for public health, usually associated with foodborne diseases. However, animal sources of these pathogens are also critical in transmission; therefore, broad research has been directed to wildlife reservoirs lately [19,20]. In this study, a community of carnivores in the Biosphere Reserve in the northwest of Mexico was analyzed to describe the genetic and virulent characteristics of *Salmonella* spp. and *E. coli* strains present in this area. Previous studies related to the composition of the carnivore population in the Janos Biosphere Reserve had reported 14 species of wild carnivores [21]. In this study, different *Salmonella* and *E. coli* strains were isolated from diverse carnivore species in the Janos Biosphere Reserve; the analysis of virulence genes, phylogroups, and serotypes showed an important diversity of these pathogens in this area (Table 2 and Table 3). Interestingly some *E. coli* serotypes were specific for each carnivore species; for example, the serogroup O153 was only found in Desert fox (*Vulpes macrotis*) and serogroup O21 in Coyote (*Canis latrans*). This phenomenon was also observed for phylogroup B2, which was only detected in Hooded skunk (*Mephitis macroura*).

The results obtained showed the existence of *Salmonella* and *E. coli* in only 4 of 55 animals captured. *Salmonella* spp. strains were recuperated from coyote (*Canis latrans*), Gray fox (*Urocyon cinereoargenteus*), Striped skunk (*Mephitis mephitis*), and Lynx (*Linx rufus*). There are several reports about the presence of *Salmonella* spp. in carnivores such as civets, otters, raccoons, foxes, coyotes, and wild felids [22,23,24,25,26,27]. In these studies, the reported frequencies range from 2.1% to 65%. The frequency of 9.09% (5/55 carnivores) in our study is much lower than the 49% reported in free-ranging raccoons in Costa Rica [28] but higher than the prevalence of 7.4% in a study of raccoons in Pennsylvania [26]. *Salmonella* sp. has been also isolated from other wild carnivores such as red fox (*Vulpes vulpes*), raccoon (*Procyon lotor*), badger (*Meles meles*), coyote (*Canis latrans*), masked palm civets (*Paguma larvata*), and otter (*Lutra lutra*) [22,26,27,29,30,31].

In this study, the *Salmonella* strains were analyzed for the presence of different virulence genes associated with intracellular invasion, survival, and bacterial adherence. The results showed only three strains positive for *inv*A and *sop*E, genes associated with the capability to invade cells. From these strains, only one was positive for all fimbrial operons evaluated in this study, and the rest have a deficiency of two operons. Various reports describe that all *Salmonella* spp. strains (from different serotypes), with positive results for the *inv*A gene, can be associated with invasive strains [17,32]. The results obtained in invasion assays with *Salmonella* spp. strains isolated from carnivores are congruent with this statement.

Previous report showed that diverse fimbriae provide to *Salmonella* Typhimurium the capability to adhere to different cell types; for example, the adhesion of this bacteria to HEp-2 cells has been proposed to be mediated by type 1 fimbriae [32,33]. However, in this study, in two *Salmonella* spp. strains, the type I fimbriae was not present; thus, the invasion of Hep2 cells by these strains should be affected. Interestingly, one of these strains (Meph 18) showed a high capability to invade Hep-2 cells, even more than reference strain *S*. *typhimurium* 14028. This result suggested that other fimbriae could participate in bacterial adhesion. Bäumler [33] reported that *lpf* fimbrial operon was enough to promote *S*. *typhimurium* adhesion to HEp-2 cells. In survival assays, only two strains showed a capability to replicate inside Hep2 cells, 1-B-Uro and Mep18A. This last strain was hyperinvasive in contrast to the 1-BR-Uro strain, which did not replicate. In other studies, *Salmonella* spp. strains isolated from reptiles showed a high capability to invade human cells [34,35]. In our study, we identified only two invasive strains. These data suggest that *Salmonella* spp. strains recovered from wildlife were able to invade human cells.

Previous studies had recognized red foxes (*Vulpes vulpes*), European badgers (*Meles meles*), and coyotes as possible reservoirs of *Salmonella* spp. [29,36,37,38]. In this study, the species with the highest frequency of *Salmonella* spp. were desert foxes, coyotes, and badgers. This result could support the hypothesis that these carnivores could be natural reservoirs for *Salmonella* spp.; however, more studies are necessary to confirm this hypothesis. Finally, cattle had also been described as a reservoir of *Salmonella* spp. [39,40]. Therefore, their presence in wild areas could result in the possible spread of *Salmonella* spp. from domestic to wild animals, altering infectious disease dynamics and creating opportunities for cross-species transmission. These phenomena carry implications for wildlife disease management and highlight areas for future work.

The presence of *E. coli* in different hosts is normal since it is a microorganism that is part of the intestinal microflora [41]. Therefore, it is not unusual to find *E. coli* in carnivores; however, for this microorganism, diverse pathogenic variants have been described, based on virulence genes, serotypes, and phylogroups. In this study these criteria were used to evaluate *E. coli* strains isolated from carnivores.

Some of the serogroups found in this study, such as O153:H7, O139:21, and O21:H21, are related to Shiga toxin-producing strains. O153:H7 has been found associated to hemorrhagic enteritis in rabbits, and some strains showed an atypical EPEC profile with the presence of *esc*V and *eae* genes. Strains O139:21 and O21:21 are also associated with enteric infections in humans and are also related with Shiga toxin-producing strains. Finally, *E.*
*coli* O17H18 strains have been associated with urinary tract infection [42].

The presence of *E. coli* strains positive for *esc*V and *eae* genes, able to produce localized adhesion and pedestal-type structure in human cells, indicates the importance of analyzing *E. coli* strains isolated from wild animals, like a potential source of virulence strains able to cause human infections. There is no clear explanation for why these pathogenic strains were found in carnivores.

The higher percentage of strains belonging to unknown phylogenetic groups found in the present study is similar to previous findings with strains retrieved from wild animals [43].

## 5. Conclusions

The distribution and incidence of emerging diseases has increased in recent years, causing economic losses and problems in public and animal health. One of the reasons for this phenomenon is the recent evolution of pathogenic microorganisms present in natural environments, with their subsequent dissemination outside their areas. In this study, the virulence characteristics of *Salmonella* spp. and *E. coli* strains isolated from carnivores in the Janos Biosphere Reserve were analyzed. The results obtained show that different species of wild carnivores can carry atypical strains of pathogenic microorganisms, suggesting the presence of potential pathogens in natural environments. On the other hand, the genetic analysis of *E. coli* strains showed that it is necessary to increase strain classification systems, since in wild animals the variations in *E. coli* may be different from those described with human strains.

## Figures and Tables

**Figure 1 animals-12-01064-f001:**
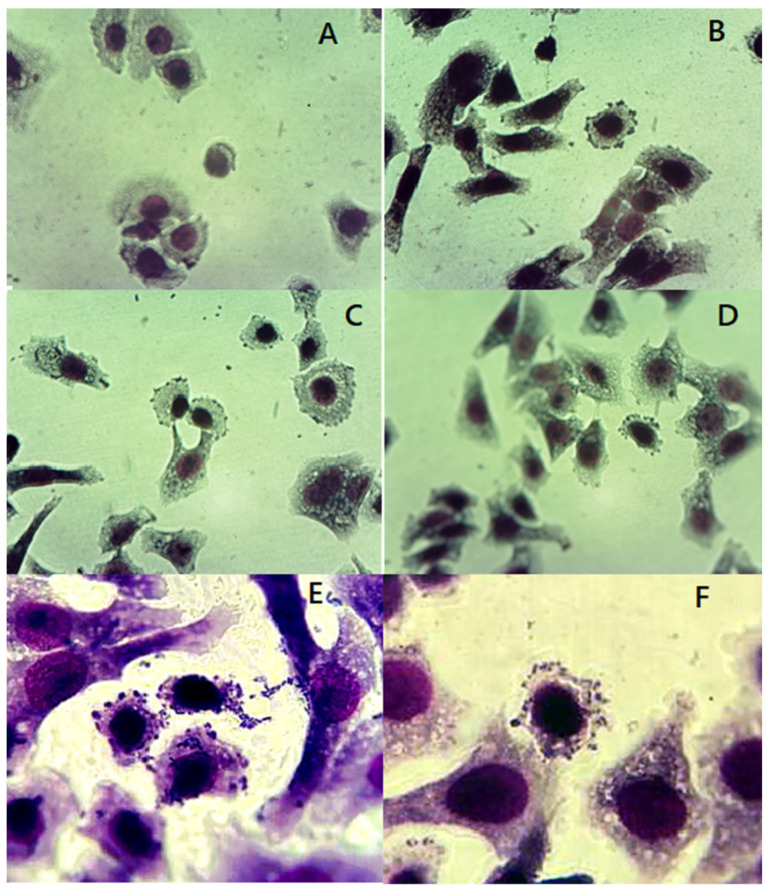
*E. coli* adhesion assays in Hep-2 cells. (**A**) Uninfected Hep-2 cells, (**B**) Hep-2 cells infected with control strain EPEC O127:H6, (**C**–**F**) *E. coli* strains *esc*V^+^, *eae*^+^ isolated from carnivores. Localized adhesions are observed, also lesions suggestive of pedestals.

**Figure 2 animals-12-01064-f002:**
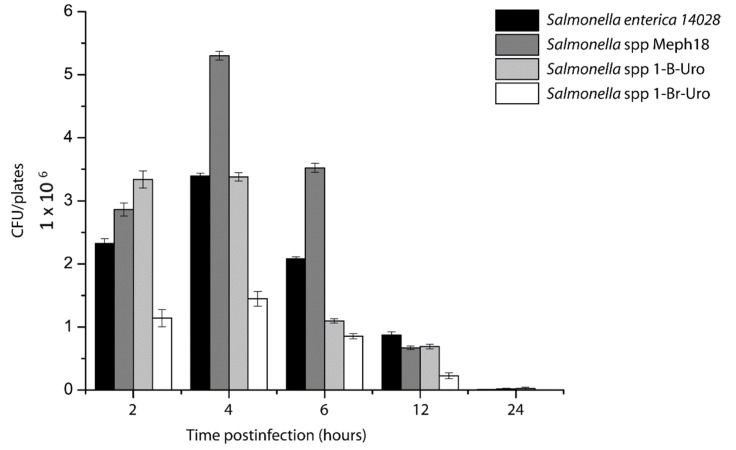
Intracellular invasion assays. Invasion capability of *Salmonella* spp. strains isolated from carnivores was determined by gentamicin-protection assays. The reference strain *S. enterica* Typhimurium 14028 and *Salmonella* spp. strains isolated from carnivores are shown.

**Table 1 animals-12-01064-t001:** *Salmonella* and *E. coli* isolates by species.

Species	Number of Animals Captured	*Salmonella* spp. Isolation	*E. coli* Isolation *
Coyote (*Canis latrans*)	15	1/15	3/15
Gray fox (*Urocyon cinereoargenteus*)	6	2/6	0/6
Desert fox (*Vulpes macrotis*)	14	0/14	2/14
Striped skunk (*Mephitis mephitis*)	2	1/2	0/2
Hooded skunk (*Mephitis macroura*)	3	0/3	2/3
Lynx (*Linx rufus*)	5	1/5	0/5
Raccoon (*Procyon lotor*)	4	0/4	0/4
Badger (*Taxidea taxus*)	6	0/6	0/6
Total animals	55	5/55 (9.09%)	7/55 (12.72%)

* Only animals with virulent *E. coli* isolations (strains with positive PCR results to virulence genes evaluated in this study) were reported. Commensal *E. coli* were found in all animals.

**Table 2 animals-12-01064-t002:** Characteristics of *Salmonella* spp. strains isolated from carnivores.

Species	Genotyping
Gray fox (*Urocyon cinereoargenteus)*	2 *Salmonella* spp. (*inv*A^+^, *sop*E^+^, *cdt*B^+^ *peg*A^+^, *std*A^+^, *lpf*A^+^, *stf*A^+^, *bcf*A^+^, *sti*A^+^, *stb*A^+^, *ste*A^+^, *fim*A^+^, *sef*A^+^)
Striped skunk (*Mephitis mephitis*)	1 *Salmonella* spp. (*inv*A^+^, *sop*E^+^, *cdt*B^+^ *peg*A^+^, *std*A^+^, *lpf*A^+^, *stf*A^+^, *bcf*A^+^, *sti*A^+^, *stb*A^+^, *ste*A^+^, *fim*A^+^, *sef*A^+^)
Coyote (*Canis latrans*)	1 *Salmonella* spp. (*inv*A^−^, *sop*E^−^, *cdt*B^−^)
Lynx (*Linx rufus*)	1 *Salmonella* spp. (*inv*A^−^, *sop*E^−^, *cdt*B^−^)

**Table 3 animals-12-01064-t003:** *E. coli* strains isolated from carnivores.

Species	Serotype	Genotyping
Desert fox (*Vulpes macrotis*)	8 *E. coli* O153H21 9 *E. coli* O153H7 1 *E. coli* O175H16 4 *E. coli* O139H21 1 *E. coli* O139 H **	6 *E. coli* (*eae*^+^) 11 *E. coli* (*esc*V^+^) 6 *E. coli* (*esc*V^+^, *eae*^+^)
Coyote (*Canis latrans*)	2 *E. coli* O75H38 1 *E. coli* O17H18 4 *E. coli* O21H21 2 *E. coli* O110H28 13 *E. coli* O ** H11	17 *E. coli* (*stx*2^+^) 5 *E. coli* (*stx*1^+^, *stx*2^+^)
Hooded skunk *(Mephitis macroura*)	7 *E. coli* O63H6 4 *E. coli* O96H49 3 *E. coli* O93H ** 2 *E. coli* nd	12 *E. coli* (*esc*V^+^) 3 *E. coli* (*stx*1^+^) 1 *E. coli* (*esc*V^+^, *eae*^+^)

** For these strains, agglutination did not allow assigning a specific group.

## Data Availability

The raw data supporting the conclusions of this manuscript will be made available by the authors, without undue reservation, to any qualified researcher.

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
