# Peer review of "Characterization of Salmonella spp. and E. coli Strains Isolated from Wild Carnivores in Janos Biosphere Reserve, Mexico"

_animals, 2022, doi:10.3390/ani12091064_

Round 1

Reviewer 1 Report

This is an interesting article which investigates E. coli and Salmonella in wild carnivores within Mexico. It is generally well written and will add to the growing area on pathogen carriage in these animals.

I have a few minor comments which are below

I have tried to offer suggestions for rewording of sections too

Please ensure that all bacterial names are in italics throughout the manuscript, and in vitro/ in vivo as well

Please also include manufacturers for all reagents

Please also ensure that gene names are in italics

It is written unknown rather than unknow throughout

Also please be consistent with the use of spp. or sp.

Line 23- whats DIDs? Maybe EIDs?

Line 54- quarantine and drug and vaccine development (reword)

Line 82- assessed in a wild population (reword)

Line 118- delete ‘A’ at the start of the sentence

Line 121-123- most of these would read better as …were positive to the XXX test’

Line 129- came instead of come

Line 129- what do you mean by enteric processes?

Line 145- this should be multiplex PCR rather than the other way round

Line 168- incubated for five hours …(Reword)

Line 192- by species

Line 204- virulence genes rather than virulent genes

Line 208- can you be more precise that practically all individuals?

Line 212- three of them showed amplification … (reword)

Line 214- codify doesn’t sound quite right. Please reword

Line 216-219- here you introduce some strains of bacteria which haven’t been mentioned before, so the reader doesn’t know much about them. Could you include some details on them?

Line 285-287- I couldn’t understand this, so please reword it

Line 303- 307- I couldn’t understand this, so please reword it

Line 330- from domestic animals to wildlife (reword)

Line 342- are associated rather than is associated

Line 342-343- this is not a sentence, so please merge with the following sentence

Line 345-347- I couldn’t understand this, so please reword it

Author Response

Author's Reply to the Review Report (Reviewer 1)

  1. Please ensure that all bacterial names are in italics throughout the manuscript, and in vitro/ in vivo as well

All bacterial names are in italics now, and “in vitro/in vivo” as well.

  1. Please also include manufacturers for all reagents.

All reagents’ manufactures were included.

  1. Please also ensure that gene names are in italics.

All gene names are in italics now.

  1. It is written unknown rather than unknow throughout Also please be consistent with the use of spp. or sp.

The word unknow was replaced, also all sp. by spp.

  1. Line 23- whats DIDs? Maybe EIDs?

I apologize, it was a mistake, DIDs was changed by EIDs.

  1. Line 54- quarantine and drug and vaccine development (reword).

The phrase was changed to “Global efforts to reduce the impacts of emerging diseases are largely focused on post-emergence outbreak control, through quarantine procedures, antimicrobial drug development, and vaccination.”

  1. Line 82- assessed in a wild population (reword).

The phrase was changed to “In this study, E. coli and Salmonella spp. strains were isolated from a wild population of carnivores from the Janos Biosphere Reserve Chihuahua, Mexico, and the virulence characteristics of strains were described”.

  1. Line 118- delete ‘A’ at the start of the sentence.

The letter “A” was deleted.

  1. Line 121-123- most of these would read better as …were positive to the XXX test’ Line 129- came instead of come.

The phrase was changed by “Thirty-six Green metallic sheen-like colonies from each sample were picked and test-ed in MRVP broth (Oxoid, USA), Simmons citrate (Oxoid, USA), SIM agar (Oxoid, USA), TSI agar (Oxoid, USA) and urea agar (Oxoid, USA). Isolates that were positive to the methyl red test (Sigma-Aldrich, USA), negative to the Voges-Prokeur test (Merck, USA), negative to the Simmons citrate test, negative to the H2S test, positive to the in-dole test (Merck, USA), positive to the motility test, negative to the urease production test and produced acid over acid reaction in the TSI test, were subcultured on EMB agar (Oxoid, USA) at 37°C for 24 hours”. Also, the word come was changed.

  1. Line 129- what do you mean by enteric processes?

I apologize, it was a mistake, “enteric processes” was changed by “Enteric infections”.

  1. Line 145- this should be multiplex PCR rather than the other way round.

We used a multiplex PCR; the redaction was changed to avoid misunderstanding.

  1. Line 168- incubated for five hours …(Reword)

The phrase was changed to “Cells were inoculated with an infection dose of 1000:1 bacterium per cell and incubated at 37°C by five hours”.

  1. Line 192- by species

The word “specie” was changed.

  1. Line 204- virulence genes rather than virulent genes

The word “virulent” was replaced by “virulence”.

  1. Line 208- can you be more precise that practically all individuals?

The word “practically” was removed

  1. Line 212- three of them showed amplification … (reword)

The phrase was changed to “All Salmonella strains showed positive results for 16s-ITS, PCR, but only three strains had positive results for invA, cdtB, and sopE gene”.

  1. Line 214- codify doesn’t sound quite right. Please reword

The phase was changed by “Considering that invA and sopE genes are essential virulence factors associated with cell invasion [17; 18], strains with negative results for these genes were not included in the analysis of fimbrial operons and invasion assays”

  1. Line 216-219- here you introduce some strains of bacteria which haven’t been mentioned before, so the reader doesn’t know much about them. Could you include some details on them?

Details from these strains were included, and Table 2 was changed to avoid misunderstanding. The phrase was replaced by “For PCR analysis of fimbrial operons, the strain 1-B-Uro (strain isolated from Gray fox) showed positive results for all evaluated genes. On the other hand, strains 1-BR-Uro (isolated from Gray fox) and Meph 18 (isolated from Striped skunk) lacked fimA and steA genes, and the strain Meph 18 was also deficient in the stfA gene (Table 2)”.

  1. Line 285-287- I couldn’t understand this, so please reword it

The text was replaced by “In this study, different Salmonella and E. coli strains were isolated from diverse carnivore species in the Janos Biosphere Reserve; the analysis of virulence genes, phylogroups, and serotypes showed an important diversity of these pathogens in this area (Table 2 and 3). Interestingly some E. coli serotypes were specific for each carnivore species; for example, the serogroup O153 was only founded in Desert fox (Vulpes macrotis) and serogroup O21 in Coyote (Canis latrans). This phenomenon was also observed for phylogroup B2, which was only detected in Hooded skunk (Mephitis macroura)”.

  1. Line 303- 307- I couldn’t understand this, so please reword it

The text was replaced by “Various reports describe that all Salmonella spp. strains (from different serotypes), with positive results for the invA gene, can be associated with invasive strains [17, 32]. The results obtained in invasion assays with Salmonella spp strains isolated from carnivores are congruent with this statement.”

  1. Line 330- from domestic animals to wildlife (reword)

The text was replaced by “Therefore, their presence in wild areas could result in the possible spread of Salmonella spp. from domestic to wild animals, altering infectious disease dynamics and creating opportunities for cross-species transmission.”

  1. Line 342- are associated rather than is associated

I apologize, it was a mistake, the singular verb was changed by plural

  1. Line 342-343- this is not a sentence, so please merge with the following sentence Line 345-347- I couldn’t understand this, so please reword it

The text was replaced by “The presence of E. coli strains positive to escV and eae genes, able to produce localized adhesion and pedestal-type structure in human cells, indicates the importance of analyzing E. coli strains isolated from wild animals, like a potential source of virulence strains able to cause human infections. There is no clear explanation for why these pathogenic strains were found in carnivores”.

Reviewer 2 Report

Manuscript ID aninals-1679472 – “Characterization of E. coli and Salmonella strains isolated from wild carnivores in Janos Biosphere Reserve, Mexico”.

In my opinion, this is an interesting manuscript that characterises E. coli and Salmonella isolates in free-living animals. It is worth noting that although the population of animals studied is not sizable, Salmonella was nevertheless found in 5 wild animals. This may also indicate human impact on nature, including free-living animals. The paper has some shortcomings to which the authors should pay attention.

It would be good to briefly describe in the introduction what the genes under study encode or you could include this information in an additional (second) column in the supplementary material described as Supplementary Table 1. Primer sequences and conditions for Salmonella and E. coli genetic characterization by PCR.

L204: I can’t see Table 4.

L227: Please correct the numbering of tables in the manuscript and supplementary materials.

The Table 4 mentioned appears to be labeled as Table 2 in the supplementary materials .

The numbers of tables, as well as figures, should be numbered according to the order of citation in the text, e.g. Figure 2 is cited first, followed by figure 1.

Ls207, 224, 234, 255 and elsewhere: E.coli and Salmonella should be written in italics, as should the gene name.

Ls249, 252 and elsewhere: Please correct notation 106.

L252: UFC or CFU?

Author Response

Author's Reply to the Review Report (Reviewer 2)

Manuscript ID aninals-1679472 – “Characterization of E. coli and Salmonella strains isolated from wild carnivores in Janos Biosphere Reserve, Mexico”.

In my opinion, this is an interesting manuscript that characterizes E. coli and Salmonella isolates in free-living animals. It is worth noting that although the population of animals studied is not sizable, Salmonella was nevertheless found in 5 wild animals. This may also indicate human impact on nature, including free-living animals. The paper has some shortcomings to which the authors should pay attention.

It would be good to briefly describe in the introduction what the genes under study encode or you could include this information in an additional (second) column in the supplementary material described as Supplementary Table 1. Primer sequences and conditions for Salmonella and E. coli genetic characterization by PCR.

The information about of each gene was included in supplementary table 1

  1. L204: I can’t see Table 4.

I apologize it was a mistake, the table number was 2, in the supplementary material

  1. L227: Please correct the numbering of tables in the manuscript and supplementary materials. The Table 4 mentioned appears to be labeled as Table 2 in the supplementary materials.

The numbering of tables was revised.

  1. The numbers of tables, as well as figures, should be numbered according to the order of citation in the text, e.g. Figure 2 is cited first, followed by figure 1.

The figures were changed to correct this mistake.

  1. Ls207, 224, 234, 255 and elsewhere: E.coli and Salmonella should be written in italics, as should the gene name.

All scientific names are in italic now.

Ls249, 252 and elsewhere: Please correct notation 106. L252: UFC or CFU?

I apologize it was a mistake, UFC was changed by CFU.

Reviewer 3 Report

The work is meaningful, well understood and scientifically correct and valid.

It's also very important to reinforce the conclusions given the richness of the data produced and the species investigated.

Author Response

Author's Reply to the Review Report (Reviewer 3)

Manuscript ID aninals-1679472 – “Characterization of E. coli and Salmonella strains isolated from wild carnivores in Janos Biosphere Reserve, Mexico”.

Comments and Suggestions for

Authors

The work is meaningful, well understood and scientifically correct and valid.

It's also very important to reinforce the conclusions given the richness of the data produced and the species investigated.

  • Thank you very much for your suggestions, they have been taken care of.
